# Learning from Loss Landscape: Generalizable Mixed-Precision Quantization via Adaptive Sharpness-Aware Gradient Aligning

Lianbo Ma [1]   Jianlun Ma [1]   Yuee Zhou [1]   Guoyang Xie [2 3]   Qiang He [4]   Zhichao Lu [2]

## Abstract

Mixed Precision Quantization (MPQ) has become an essential technique for optimizing neural network by determining the optimal bitwidth per layer. Existing MPQ methods, however, face a major hurdle: they require a computationally expensive search for quantization policies on large-scale datasets. To resolve this issue, we introduce a novel approach that first searches for quantization policies on small datasets and then generalizes them to large-scale datasets. This approach simplifies the process, eliminating the need for large-scale quantization fine-tuning and only necessitating model weight adjustment. Our method is characterized by three key techniques: sharpness-aware minimization for enhanced quantization generalization, implicit gradient direction alignment to handle gradient conflicts among different optimization objectives, and an adaptive perturbation radius to accelerate optimization. Both theoretical analysis and experimental results validate our approach. Using the CIFAR10 dataset (just 0.5% the size of ImageNet training data) for MPQ policy search, we achieved equivalent accuracy on ImageNet with a significantly lower computational cost, while improving efficiency by up to 150% over the baselines.

## 1. Introduction

With the fast development of edge intelligence applications, it is desirable to compress DNNs with minimal/no performance deterioration according to specific hardware configurations (Shuvo et al., 2022; Ma et al., 2024). To realize this,

one effective way is to utilize a set of small bitwidths to quantize the entire network for reducing model redundancy (Deng et al., 2020), termed as mixed-precision quantization (MPQ). Different from fixed-precision quantization (FPQ) (Esser et al., 2019), MPQ works in a fine-grained way, allowing different bitwidths for different layers (Elthakeb et al., 2020; Guo et al., 2020). That is, the quantization-insensitive layers can be quantized using much smaller bitwidths than the quantization-sensitive layers, which can naturally obtain more optimal accuracy-complexity trade-off than FPQ.

**Limitations.** Most existing MPQ methods require **the consistency of datasets** for bitwidth search and network deployment to guarantee policy optimality, which results in **heavy search burden** on large-scale datasets (Zhao et al., 2024). For example, HAQ (Wang et al., 2019) requires approximately 72 GPU hours to search for the optimal quantization policy for ResNet50 on ImageNet. A natural solution to solve the above issue is to decouple the dataset used in MPQ search stage and model inference stage. In this way, the search efficiency can be largely improved since the MPQ search can be performed with a small size of proxy dataset. Unfortunately, it inevitably suffers from intractable challenges incurred by shifted data distributions and small data volume of the disparate datasets. For example, when CIFAR10 is used to search for MPQ policy for MobileNet-V2 trained on ImageNet, PACT (Choi et al., 2018) suffers from a substantial loss of nearly 10% in Top-1 accuracy.

To bridge such gap, we aim to search transferable policy using small proxy datasets via enhancing generalization of quantization search (as shown in Figure 7 in Supplementary Material A.1). Several studies demonstrate that utilizing the discriminative nature of feature representations (e.g., the attribution rank of image's features (Wang et al., 2021), the class-level information of feature maps (Tang et al., 2023)) can find a transferable policy. However, such disentangled representation learning methods just enhance the discrimination ability of target quanitzed models, which is empirically helpful to improve generalization, but still suffers from lack of theoretical support. Moreover, they entail intricate calculation of feature maps (e.g., attribution rank calculation and alignment, classification margin computation), which is computationally complex.

---

[1]College of Software , Northeastern University, Shenyang, China [2]The Department of Computer Science, City University of Hong Kong, Hong Kong, China [3]The Department of Intelligent Manufacturing, CATL, Ningde, China [4]College of Computer Science and Engineering, Northeastern University, Shenyang, China. Correspondence to: Guoyang Xie <guoyang.xie@ieee.org>.

*Proceedings of the 42nd International Conference on Machine Learning*, Vancouver, Canada. PMLR 267, 2025. Copyright 2025 by the author(s).

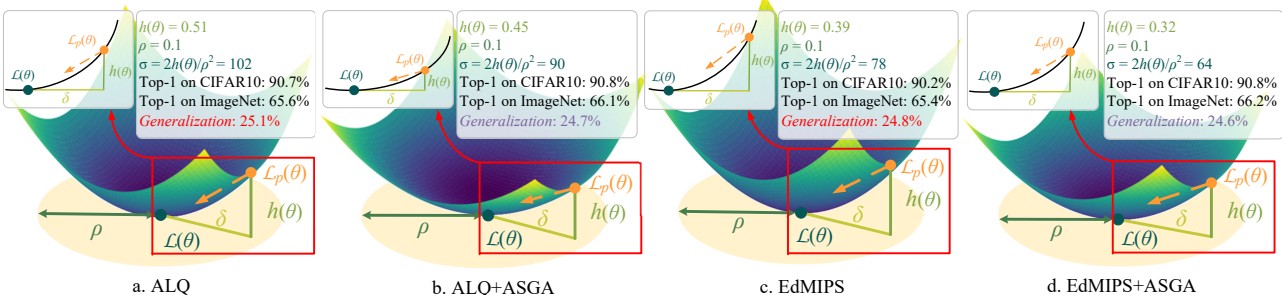

Figure 1. Comparison of the generalization performance between the baseline MPQ methods and ASGA on ResNet18 with CIFAR10. $\sigma$ serves as a measure of the sharpness of the loss landscape (defined in Section 2.2) and *Generalization* means the change in terms of Top-1 accuracy of the model on CIFAR10 and ImageNet. Compared to the baselines (a and c), ASGA significantly reduces surrogate gap (i.e., the difference between the perturbed loss $\mathcal{L}_p(\theta)$ and the experience loss $\mathcal{L}(\theta)$), smoothing the sharpness of the loss landscape, thereby enhancing the model's generalization on the target dataset (b and d).

**Motivation.** Our idea is motivated by the observation that there exists the strong relationship between the sharpness of loss landscape in MPQ training and the generalization of target quantized model over unseen datasets (see Figure 1). Moreover, the quantization noise could exacerbate the sharpness of loss curvature and blur the minimum of loss landscape. Since **minimizing the sharpness** of the loss landscape is significantly conducive to generalization enhancement, we seek the MPQ policy that can lead to flatter loss minima of quantized model, making the loss landscape more robust to quantization noise. Note that the calculation of sharpness is relatively simple and cheap with no need of intricate computation of feature maps.

**Contributions.** In this paper, we present an **a**daptive **s**harpness-aware **g**radient **a**ligning (**ASGA**) method to learn generalizable MPQ policy, which exploits the sharpness information of loss landscape for generalization improvement on the proxy datasets, as shown in Figure 2. To incorporate the loss sharpness into the optimization of MPQ training, we handle gradient conflicts between different optimization objectives by implicitly aligning gradient directions, and then accelerate the optimization process by setting an adaptive perturbation radius. To the best of out knowledge, our design is **the first attempt** to apply **sharpness-based generalization** to MPQ policy search, which offers advantages such as no intricate computation of feature maps and high search efficiency.

Experimental results demonstrate that a small sharpness measure learned from proxy dataset helps seek a transferable MPQ policy for quantizing the model trained on large datasets. Our method acquires promising performance when searching on very small proxy datasets versus directly on large datasets, where the size of the former is only 0.5% of the latter. As a result, we accomplish impressive improvement of MPQ search efficiency. For ResNet18, ResNet50 and MobileNet-V2, by using CIFAR10 as the proxy dataset,

our method gets 150%, 127% and 113% speedup compared to state-of-the-art (SOTA) MPQs, respectively.

## 2. Approach

### 2.1. Problem Formulation

**General Formulation.** The main goal of MPQ is to quantize the model using suitable bitwidths from the quantization policy $\mathcal{Q}$ under resource-constrained conditions $\Omega_0$, which can be formulated as the following bi-level optimization problem:

$$\min_{\mathcal{Q}} \mathcal{L}_{val}\left(\boldsymbol{\theta}^*, \mathcal{Q}\right),$$
$$\text{s.t. } \boldsymbol{\theta}^* = \arg\min \mathcal{L}_{train}(\boldsymbol{\theta}, \mathcal{Q}), \ \Omega(\mathcal{Q}) \leqslant \Omega_0, \quad (1)$$

where $\mathcal{L}_{val}$ and $\mathcal{L}_{train}$ represent the task loss on the validation dataset and the training dataset, respectively, and $\boldsymbol{\theta}^*$ is the optimal weights set of quantized network under $\mathcal{Q}$. By alternatively updating $\mathcal{Q}$ and $\boldsymbol{\theta}$ until convergence, we can obtain the optimal mixed-precision model.

Due to differences in data distribution, most existing MPQ methods need consistency between the datasets used in policy search stage and the dataset used in model inference stage, which requires significant computation costs in large datasets. It is also fragile in privacy-sensitive MPQ tasks, where the training data is not allowed to directly come from the validation dataset. To break above limitations, we aim to directly learn optimal MPQ policy on small proxy dataset and then generalize it to target large dataset for inference, and then reformulate the search objective of MPQ as:

$$\min_{\mathcal{Q}} \mathbb{E}\left(\mathcal{L}_{val}\left(\boldsymbol{\theta}^*, \mathcal{Q}, \boldsymbol{x}\right)\right), \boldsymbol{x} \in \mathcal{D}_{val},$$
$$\text{s.t. } \boldsymbol{\theta}^* = \arg\min \mathbb{E}\left(\mathcal{L}_{train}\left(\boldsymbol{\theta}, \mathcal{Q}, \boldsymbol{x}\right)\right), \quad (2)$$
$$\boldsymbol{x} \in \mathcal{D}_{train-proxy}, \Omega(\mathcal{Q}) \leqslant \Omega_0,$$

where $\mathbb{E}$ denotes the expectation, $\mathcal{D}_{val}$ is the dataset containing all validation data in model inference stage and

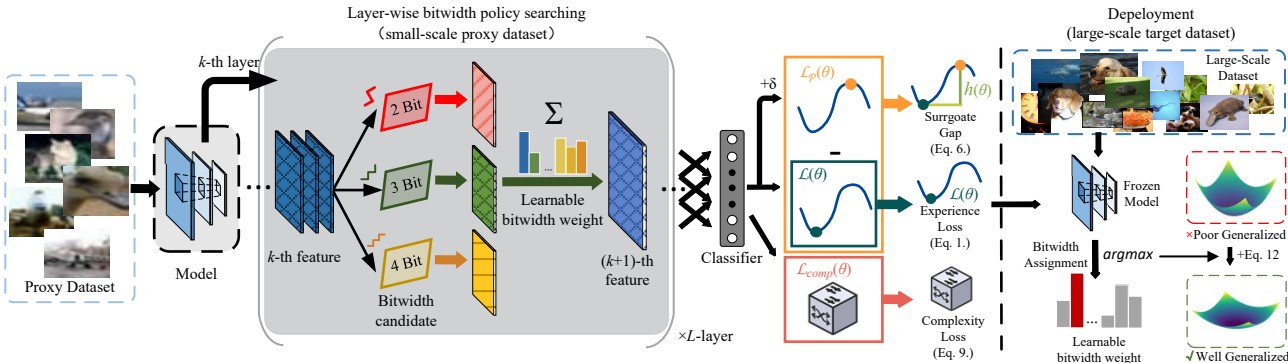

Figure 2. The illustration of our approach. We aim to search for an optimal quantization policy with a flat loss landscape on a proxy dataset, which can be applied to large-scale target datasets. In the policy searching stage, we seek a MPQ policy with a flat loss landscape by minimizing the empirical loss, complexity loss, and surrogate gap. In the deployment stage, the searched MPQ policy can be directly applied to model inference on large target datasets.

$\mathcal{D}_{train-proxy}$ is the proxy dataset in MPQ search stage.

**Differentiable Formulation.** We consider a differentiable MPQ (DMPQ) search process (Cai & Vasconcelos, 2020; Yu et al., 2020), which formulates the whole searching space as a supernet (as Directed Acyclic Graph) where the nodes represent candidate quantization bitwidths, and the edges denote learnable weights of the bitwidths. The optimization objective of DMPQ is defined as

$$\mathcal{L}_{train}(\theta) = \mathcal{L}(\theta) + \lambda \mathcal{L}_{comp}(\theta), \quad (3)$$

where $\lambda$ is a coefficient controlling the accuracy-complexity trade-off, $\mathcal{L}(\theta)$ denotes the accuracy loss, and $\mathcal{L}_{comp}(\theta)$ is the loss of target model's complexity, i.e.,

$$\mathcal{L}_{comp}(\theta) = \sum_{l=0}^{L} \left( \sum_{j=0}^{\|B^\theta\|} \left( p_j^{l,\theta} b_j^\theta \right) \sum_{k=0}^{\|B^a\|} \left( p_k^{l,a} b_k^a \right) \right) comp^l,$$

$$s.t. \quad p_j^{l,\theta} = \frac{\exp\left(\alpha_j^l\right)}{\sum_{k=0}^{\|B^\theta\|} \exp\left(\alpha_k^l\right)}, \quad p_k^{l,a} = \frac{\exp\left(\beta_k^l\right)}{\sum_{k=0}^{\|B^a\|} \exp\left(\beta_k^l\right)},$$
$$(4)$$

where $B^\theta$ and $B^a$ denote the candidate bitwidth sets for weights and activations, respectively, $\alpha^l$ and $\beta^l$ denote the learnable weights vectors of their corresponding bitwidth candidates in layer $l$, and $comp^l$ is the BOPs (Billions of Operations Per Second) constraint of layer $l$, i.e.,

$$comp^l = c_{in}^l \times c_{out}^l \times k_a^l \times k_b^l \times h_{out}^l \times w_{out}^l, \quad (5)$$

where $c_{in}$ and $c_{out}$ denote the number of input channels and output channels, respectively, $k_a$ and $k_b$ are the kernel size, $w_{out}$ and $h_{out}$ denote the width and height of the output feature map, respectively.

One may argue that we can directly utilize subset of target dataset to search MPQ policy rather than proxy dataset transfer, but this would result in serious performance deterioration, as demonstrated in Section 3.3.3.

## 2.2. Exploiting the Loss-Sharpness Information

From the perspective of the loss landscape information in a well-preforming MPQ policy, their sharpness in MPQ training should be minimized, which has the following benefits: a) It helps the training of quantization to escape sharp region, and also alleviates the sharpness aggravation incurred by quantization noise as well. b) It has a desirable and dataset-independent property, as recent research (Foret et al., 2020; Wang et al., 2023) recognizes a small loss sharpness contributes to model generalization. c) The calculation cost of sharpness measure is relatively simple and cheap.

Motivated by this, we aim to search the MPQ policy that guarantees the loss landscape of the proxy data distribution as flat as possible. As discussed above, such a general property in MPQ search can ensure usability across the data distributions. However, the sharpness of the loss landscape is not explicitly incorporated in the current cross-entropy loss formulation. Consequently, the objective is not only to optimize accuracy and complexity, but also to seek an MPQ policy that minimizes the sharpness of the loss landscape.

**Preliminary.** In conventional model training, **S**harpness-**A**ware **M**inimization (SAM) (Foret et al., 2020) aims to search for a flatter region around the minimum with low loss values. To realize this, SAM introduces perturbations $\delta$ to model weights $\theta$, and solves the following min-max optimization optimization problems:

$$\min_\theta \mathcal{L}_p(\theta),$$
$$s.t. \ \mathcal{L}_p(\theta) \triangleq \max_{\|\delta\|_2 \le \rho} \mathcal{L}(\theta + \delta), \quad (6)$$
$$\delta \sim \mathcal{N}\left(0, b^2 I^k\right), \delta \in \mathbb{R}^k,$$

where $\rho$ is the perturbation radius, $k$ is the dimension of $\theta$ and $b$ is the scaling factor influenced by the candidate bitwidth. For a well-trained model, when the perturbed loss is minimized, the neighborhood corresponds to low losses

(below the perturbed loss). Given $\rho$, through Taylor expansion around $\theta$, the inner maximization of Eq. (6) becomes a linear constrained optimization with solution:

$$
\begin{aligned}
&\arg\max \mathcal{L}_p(\theta) \\
&= \arg\max \mathcal{L}(\theta) + \delta^\top \nabla\mathcal{L}(\theta) + o\left(\rho^2\right) \\
&\approx \rho \frac{\nabla\mathcal{L}(\theta)}{\|\nabla\mathcal{L}(\theta)\|}.
\end{aligned}
\tag{7}
$$

Then, the optimization problem of SAM reduces to:

$$
\begin{aligned}
&\min_\theta \mathcal{L}_p(\theta) \approx \min_\theta \mathcal{L}\left(\theta_p\right), \\
&\text{where } \theta_p \triangleq \theta + \rho \frac{\nabla\mathcal{L}(\theta)}{\|\nabla\mathcal{L}(\theta)\|},
\end{aligned}
\tag{8}
$$

where $\theta_p$ denotes the perturbation weights corresponding to the maximum perturbation loss within the neighborhood. In this way, the goal of SAM is converted to seek an optimal solution on the perturbed loss landscape of $\mathcal{L}_p(\theta)$.

**Adaptive Sharpness-Aware Gradient Aligning.** SAM suffers from following drawbacks when applied in MPQ:

(1) For a fixed $\rho$, there is no linear relationship between the perturbed loss $\mathcal{L}_p(\theta)$ and the sharpness loss $\sigma_{max}$ (the dominant eigenvalue of the Hessian). Here, $\boxed{\sigma_{\max} \approx \frac{2h(\theta)}{\rho^2}}$, where $\boxed{h(\theta) = \mathcal{L}_p(\theta) - \mathcal{L}(\theta)}$ denotes the surrogate gap.

We can see a smaller $\mathcal{L}_p(\theta)$ does not necessarily guarantee convergence to a flat region, as shown in Figure 3. When $\mathcal{L}(\theta_1) < \mathcal{L}(\theta_2)$, even if $\mathcal{L}_p(\theta_1)$ is much smaller than $\mathcal{L}_p(\theta_2)$, SAM may still favor the minimum on the left side.

(2) When optimizing $\mathcal{L}_p(\theta)$ and $h(\theta)$ simultaneously, there exists conflict between $\nabla\mathcal{L}(\theta)$ and $\nabla\mathcal{L}_p(\theta)$ (Zhuang et al.). This is because when minimizing $h(\theta)$, the orthogonal component $\nabla\mathcal{L}(\theta)_\perp$ will increase $\mathcal{L}(\theta)$, thereby making the decrease of $\mathcal{L}_p(\theta)$ difficult since $h(\theta)$ is always non-negative, which hurts the model's generalization.

(3) Since the loss landscape evolves towards being flatter as the search progresses, it is hard to capture surrogate gap by fixed $\rho$, slowing down the speed of optimizing sharpness.

Fortunately, from basic principles of vector operations, we find that when $\nabla\mathcal{L}(\theta)$ and $\nabla\mathcal{L}_p(\theta)$ are consistent, $\nabla h(\theta) = \nabla\mathcal{L}_p(\theta) - \nabla\mathcal{L}(\theta)$ is also consistent with them, which makes it feasible to apply the gradient descent to all three losses effectively [1]. Hence, as shown in Eq. (9), we reformulate the optimization objective of MPQ as seeking a balance between $\mathcal{L}(\theta)$, $\mathcal{L}_p(\theta)$, and $h(\theta)$ by implicitly aligning the gradient directions between $\mathcal{L}(\theta)$ and $\mathcal{L}_p(\theta)$ while minimizing $\mathcal{L}(\theta)$,

---

[1]Regarding to this point, we provide more explanations in Supplementary Material A.2.1.

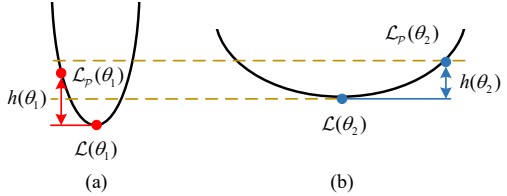

Figure 3. (b) exhibits a flatter landscape compared to (a), as indicated by its smaller $h(\theta_2)$. Nevertheless, SAM favors (a) due to its lower local minimum $\mathcal{L}(\theta_1)$.

$\mathcal{L}_p(\theta)$ and the angles between their gradients:

$$
\begin{aligned}
&\min_\theta(\mathcal{L}(\theta), \mathcal{L}_p(\theta), h(\theta)) \\
&= \min_\theta(\mathcal{L}(\theta), \mathcal{L}_p(\theta - \mu\nabla\mathcal{L}(\theta))),
\end{aligned}
\tag{9}
$$

where $\mu$ is the scaling factor. In this way, the training loss can converge to a flat region with a small loss value. More details of gradient alignment are provided in Supplementary Material A.2.2.

Then, we suggest an adaptive strategy to gradually adjust $\rho$ according to $h(\theta)$, i.e., $\rho = \min(\rho_{max}, \frac{\phi}{\ln(h(\theta)+1)})$, aiming to obtain optimal gradient direction for sharpness optimization. By integrating above strategies, we get the design of adaptive sharpness-aware gradient aligning (ASGA):

$$
\begin{aligned}
&\min_\theta(\mathcal{L}(\theta), \mathcal{L}_p(\theta), h(\theta)) \\
&= \min_\theta(\mathcal{L}(\theta), \mathcal{L}(\theta + (\frac{\rho}{\|\nabla\mathcal{L}(\theta)\|} - \mu)\nabla\mathcal{L}(\theta))), \\
&s.t.\ \rho = \min(\rho_{max}, \frac{\phi}{\ln(h(\theta)+1)}).
\end{aligned}
\tag{10}
$$

More details about the merits of our design are provided in Supplementary Material A.3. Note that we do not need to recalculate $\mathcal{L}(\theta)$, which has been determined when calculating $\nabla\mathcal{L}(\theta)$, and thus ASGA has an equivalent computation complexity to SAM.

### 2.3. Generalizable MPQ via Adaptive Sharpness-Aware Gradient Aligning

Then, we add the optimization objective of our proposed ASGA into the loss design of DMPQ as the regularization of quantization generalization, i.e.,

$$
\min_\theta \left(\mathcal{L}(\theta), \lambda\mathcal{L}_{comp}(\theta), \mathcal{L}_p(\theta), h(\theta)\right).
\tag{11}
$$

Using Eq. (10), the above optimization objective becomes:

$$
\min_\theta(\mathcal{L}(\theta), \lambda\mathcal{L}_{comp}(\theta), \epsilon\underbrace{\mathcal{L}(\theta + (\frac{\rho}{\|\nabla\mathcal{L}(\theta)\|} - \mu)\nabla\mathcal{L}(\theta))}_{\text{the corresponding loss}})),
\tag{12}
$$

where $\epsilon$ denotes the hyper-parameters for weighting the corresponding loss in the optimization process.

In essence, the goal of our method is to seek a balance between prediction accuracy (via minimizing $\mathcal{L}(\theta)$ and $\mathcal{L}_{\mathrm{p}}(\theta)$), model complexity (via minimizing $\mathcal{L}_{comp}(\theta)$) and generalization capability (via minimizing $h(\theta)$).

## 2.4. Theoretical Analysis

### 2.4.1. GENERALIZATION PERFORMANCE ANALYSIS

First, we theoretically analyze the impact of loss sharpness on the upper bound of quantization generalization error.

---

**Lemma 1.** *Suppose the training set contains $m$ elements drawn i.i.d. from the true distribution and the average loss in MPQ search is $\mathcal{L}(\theta) = \frac{1}{m}\sum_{i=1}^{m}\mathcal{L}_i(\theta, x_i, y_i)$ ($\mathcal{L}_i(\theta)$ for short), where $(x_i, y_i)$ is the input-target pair of $i$-th element. For the prior distribution (independent of training) $\zeta$, we have:*

$$\mathbb{E}_{\theta \sim \tau}\mathbb{E}_{x_i}\mathcal{L}_i(\theta) \leq \mathcal{L}_p(\theta) + \mathcal{R}, \quad (13)$$

*with probability at least $(1-a)\left[1 - e^{-\left(\frac{\rho}{\sqrt{2}b} - \sqrt{k}\right)^2}\right]$, where $a \in (0,1)$ denotes confidence level, and $\mathcal{R} = 4\sqrt{\left(KL(\tau\|\zeta) + \log\frac{2m}{a}\right)/m}$. This shows that ASGA effectively reduces the upper bound of the generalization error during the MPQ process.*

---

**Proof.** The detailed proof is provided in Supplementary Material A.4 .

### 2.4.2. CONVERGENCE ANALYSIS

Then, we analyze the convergence property of ASGA-based DMPQ under non-convex setting, as shown below.

---

**Lemma 2.** Under the condition of employing SGD with a learning rate of $\gamma = \frac{\gamma_0}{\sqrt{T}} \leq \frac{1}{\beta}$ as the base optimizer, we have:

$$\sum_{t=1}^{T}\mathbb{E}\|\nabla\mathcal{L}(\theta_t)\|^2 \leq \frac{2T\Delta}{\gamma_0\Lambda} + \frac{\Theta}{2\Lambda}, \quad (14)$$

where $\Delta = \mathbb{E}[\mathcal{L}(\theta_1) - \mathcal{L}(\theta^*)]$ ($\theta^*$ is the optimal solution), $\Theta = \rho^2\beta^2(3\beta\gamma_0 - \sqrt{T}) + \gamma_0\beta\mathcal{M}$ and $\Lambda = 3\sqrt{T} - 2\beta\gamma_0$.

---

**Proof.** We follow two basic assumptions widely used in convergence analysis of stochastic optimization (Jiang et al., 2023). **Assumption 1:** Suppose there exists a constant $\beta > 0$ such that for all $\theta, v \in \mathbb{R}^d$, $\frac{1}{\beta}|\nabla L(\theta) - \nabla L(v)\|_2 \leq \|\theta - v\|_2$ holds. **Assumption 2:** For any data batch $\mathcal{B}$, there exists a positive constant $\mathcal{M}$ such that the variance of the

gradient estimator $\nabla L_{\mathcal{B}}(\theta) - \nabla L(\theta)$ is bounded by $\mathcal{M}$ for all $\theta \in \mathbb{R}^d$, i.e., $\mathbb{E}\left[\|\nabla L_{\mathcal{B}}(\theta) - \nabla L(\theta)\|_2^2\right] \leq \mathcal{M}$.

From Assumption 1, with the definition of perturbed weight (Eq. (8)), we can derive the following expression via Taylor expansion:

$$\mathcal{L}(\theta_{t+1}) \leq \mathcal{L}(\theta_t) + \nabla\mathcal{L}(\theta_t)^\top(\theta_{t+1} - \theta_t) + \frac{\beta}{2}\|\theta_{t+1} - \theta_t\|^2$$

$$= \mathcal{L}(\theta_t) - \gamma\mathcal{G} + \frac{\gamma^2\beta}{2}(\mathcal{H} - \|\nabla\mathcal{L}(\theta_t)\|^2 + 2\mathcal{G}),$$

$$(15)$$

where $\mathcal{H} = \|\nabla\mathcal{L}_{\mathcal{B}}(\theta_p) - \nabla\mathcal{L}(\theta_t)\|^2$ and $\mathcal{G} = \nabla\mathcal{L}(\theta_t)^\top\nabla L_{\mathcal{B}}(\theta_p)$ for short, then we have:

$$\mathcal{L}(\theta_{t+1}) \leq \mathcal{L}(\theta_t) - \frac{\gamma^2\beta}{2}\|\nabla\mathcal{L}(\theta_t)\|^2$$

$$+ \frac{\gamma^2\beta}{2}\mathcal{H} - (1 - \gamma\beta)\gamma\mathcal{G}. \quad (16)$$

According to the fact that $\frac{1}{2}\|a - b\|^2 \leq \|a - c\|^2 + \|c - b\|^2$ and Assumption 2, by taking the expectation on both sides of the inequality simultaneously, we have:

$$\mathbb{E}\mathcal{L}(\theta_{t+1}) \leq \mathbb{E}\mathcal{L}(\theta_t) - \frac{\gamma^2\beta}{2}\mathbb{E}\|\nabla\mathcal{L}(\theta_t)\|^2$$

$$- (1 - \gamma\beta)\gamma\mathbb{E}\mathcal{G} + \gamma^2\beta^3\rho^2 + \gamma^2\beta\mathcal{M}. \quad (17)$$

We set a cancellation term for the last term of the above equation. By using the Cauchy-Schwaz inequality and reorganizing the Eq. (17) we have:

$$\mathbb{E}\mathcal{L}(\theta_{t+1}) \leq \mathbb{E}\mathcal{L}(\theta_t) + (\beta\gamma^2 - \frac{3}{2}\gamma)\|\nabla\mathcal{L}(\theta_t)\|^2$$

$$+ \frac{1}{2}\rho^2\beta^2\gamma(3\gamma\beta - 1) + \gamma^2\beta\mathcal{M}. \quad (18)$$

By summing Eq. (18) over $T$ iterations, we can obtain:

$$\sum_{t=1}^{T}\mathbb{E}\|\nabla\mathcal{L}(\theta_t)\|^2 \leq \frac{2T\mathbb{E}(\mathcal{L}(\theta_1) - \mathcal{L}(\theta^*))}{(3\sqrt{T} - 2\beta\gamma_0)\gamma_0}$$

$$+ \frac{\rho^2\beta^2(3\beta\gamma_0 - \sqrt{T}) + \gamma_0\beta\mathcal{M}}{6\sqrt{T} - 4\beta\gamma_0}. \quad (19)$$

From Eq. (19), we can infer that Eq. (14) holds, i.e., Lemma 2 is proven.

## 3. Experiments

### 3.1. Datasets and Implementation Details

**Datasets.** The proxy datasets $\mathcal{D}_{train-proxy}$ for MPQ search include CIFAR10 (Krizhevsky et al., 2009), Flowers (Nilsback & Zisserman, 2008), and Food (Bossard et al., 2014).

For image classification, the target large dataset $\mathcal{D}_{val}$ for model inference is ImageNet. For object detection, the target dataset is VOC (Everingham et al., 2010).

We search for the MPQ policy using the training samples of small proxy datasets. Afterwards, we quantize and fine-tune the model with the searched policy on the target large datasets, where the basic data augmentation methods are adopted. The final performance of the model is then evaluated on the ImageNet and VOC validation sets.

**Models.** We employ four representative network architectures for experiments, including: (1) ResNet-18, 50 (He et al., 2016), MobileNet-V2 (Sandler et al., 2018) for image classification, and (2) ResNet-18 and VGG16 (Simonyan & Zisserman, 2014) for object detection.

**Parameters.** For MobileNet-V2 and VGG16, the bitwidth candidates of weights and activations are $B^{\theta} = B^{a} = \{2, 3, 4, 5, 6\}$. For ResNet, the candidate bitwidths are $B^{\theta} = B^{a} = \{2, 3, 4, 6\}$. Following the previous arts (Esser et al., 2019; Wang et al., 2019), the first and last layers are fixed to 8 bits.

For searching, we set the initial perturbation radius as $\rho_0 = 0.1$, and finetune it in the ablation study. We finetune the hyper-parameter $\lambda$ in Eq. (3) according to prior works on DMPQ (Cai & Vasconcelos, 2020; Wang et al., 2021), where a higher $\lambda$ value indicates a less computation complexity policy to seek.

For finetuning (quantizing), we follow the basic quantization-aware training settings in ALQ (Qu et al., 2020) and EWGS (Lee et al., 2021).

More implementation details are provided in Supplementary Material A.5.

### 3.2. Comparison with State-Of-The-Art

We compare our ASGA with the SOTA quantization methods on the classification task, including ALQ (Qu et al., 2020), EWGS (Lee et al., 2021), HAWQ (Dong et al., 2019), HAQ (Wang et al., 2019), DQ (Uhlich et al., 2019), HMQ (Habi et al., 2020), GMPQ (Wang et al., 2021) and EdMIPS (Cai & Vasconcelos, 2020) .

#### 3.2.1. RESULTS ON IMAGENET

**ResNet.** The comparison results for ResNet-{18, 50} on ImageNet are listed in Table 1. For ResNet18, under mixed 3-bits BOPs constraints, our ASGA consistently achieves higher accuracy (in terms of both Top-1 and Top-5) than the baseline methods, while it obtains about 1.5X policy search speedup compared to EWGS, showing its advantage in reducing search costs. Especially, ASGA finds the optimal MPQ policy using only 8189 training samples, which may be owing to the small data amounts of Flowers.

Table 1. Accuracy and efficiency results for ResNet and MobileNet-V2. "Top-1/5" represents the Top-1/5 accuracy of quantized model. W/A denotes the bitwidths of weights and activations. "MP" means mixed-precision quantization. "Con-epoch" denotes the number of epochs used for convergence. (·) + A denotes the baseline method with ASGA.

| Methods | W/A | con-epoch | Top-1 | Top-5 |
|---|---|---|---|---|
| ResNet18 | | | | |
| ALQ | 3/3 | 115 | 65.6 | 87.0 |
| A+A | 3/3 | 92(23↓) | 66.1(0.6↑) | 87.1(0.1↑) |
| EWGS | 3/3 | 123 | 53.85 | 77.9 |
| E+A | 3/3 | 98(25↓) | 53.86(0.01↑) | 78.12(0.22↑) |
| EdMIPS | 3MP/3MP | 110 | 66.4 | 86.1 |
| E+A | 3MP/3MP | 75(35↓) | 67.9(1.5↑) | 87.7(1.6↑) |
| GMPQ | 3MP/3MP | 96 | 66.3 | 85.4 |
| G+A | 3MP/3MP | 80(16↓) | 66.4(0.1↑) | 86.1(0.7↑) |
| SEAM* | 3MP/3MP | 96 | 65.1 | - |
| S*+A | 3MP/3MP | 85(11↓) | 65.8(0.7↑) | - |
| ResNet50 | | | | |
| HAWQ | 4MP/4MP | 98 | 74.6 | - |
| H+A | 4MP/4MP | 92(6↓) | 74.9(0.3↑) | - |
| HAQ | 4MP/8 | 125 | 72.6 | 91.4 |
| HAQ+A | 4MP/8 | 101(24↓) | 73.5(0.9↑) | 91.6(0.2↑) |
| EdMIPS | 4MP/4MP | 119 | 71.1 | 90.4 |
| E+A | 4MP/4MP | 108(11↓) | 71.9(0.8↑) | 91.0(0.6↑) |
| GMPQ | 4MP/4MP | 104 | 71.0 | 90.1 |
| G+A | 4MP/4MP | 82(22↓) | 71.5(0.5↑) | 90.9(0.8↑) |
| MobileNet-V2 | | | | |
| DQ | 4MP/4MP | 94 | 65.1 | - |
| D+A | 4MP/4MP | 87(7↓) | 65.3(0.2↑) | - |
| HMQ | 4MP/4MP | 102 | 68.9 | - |
| H+A | 4MP/4MP | 90(12↓) | 69.1(0.2↑) | - |
| GMPQ | 4MP/4MP | 100 | 68.8 | 87.4 |
| G+A | 4MP/4MP | 91(9↓) | 69.1(0.3↑) | 88.4(1.0↑) |

* SEAM: reproduction of unpublished code based on (Tang et al., 2023)

For ResNet50, we can observe that ASGA achieves best trade-off between prediction accuracy and search efficiency. That is, ASGA acquires quite similar accuracy performance, but using much less search time (e.g., than gradient-based methods HAWQ and HAQ). These results again demonstrate the effectiveness of our method in enhancing quantization generalization.

**MobileNet-V2.** Table 1 also reports the results on MobileNet-V2 under mixed 4-bits level. For mixed 4-bits results on CIFAR10, it can be found that ASGA performs better than all the compared SOTA mixed-precision methods (DQ, HMQ and GMPQ). More specifically, our ASGA arises a 0.2% absolute gain on Top-1 accuracy over HMQ, and 0.3% higher accuracy than GMPQ. For mixed 4-bits searched on CIFAR10, ASGA obtains up to 1.13X searching efficiency improvement over HMQ.

#### 3.2.2. RESULTS ON VOC

**VGG16.** We adopt the SSD detection framework with VGG16 backbone to evaluate ASGA in object detection task. Table 2 reports the results of VGG16 without and with ASGA over multiple baseline methods. For the the accuracy-complexity trade-off, our ASGA performs powerfully, achieving better accuracy results with less search cost

Figure 4. Comparison of Top-1 accuracy (– · –) and convergence epochs (– · –) of ResNet18, ResNet50, and MobileNet-V2 on CIFAR10 with different $\rho$. The result shows that adaptive $\rho$ can significantly reduce the search cost without performance deterioration.

Table 2. The mAP (%) on VOC of VGG16 and ResNet18. W/A denotes the bitwidths of weights and activations. "MP" means mixed-precision quantization. "Con-epoch" represents the epochs used by the method to achieve convergence. Param. means the model storage cost, and $(\cdot)$ + A denotes the baseline method with ASGA.

| Methods | W/A | Param. | con-epoch | mAP |
|---|---|---|---|---|
| VGG16 with SSD | | | | |
| HAQ | 4MP/8 | 47.6 | 87 | 69.4 |
| H+A | 4MP/8 | 46.2(1.4↓) | 79(8↓) | 69.4(-) |
| EdMIPS | 4MP/4MP | 39.4 | 94 | 67.7 |
| E+A | 4MP/4MP | 39.1(0.3↓) | 73(21↓) | 67.8(0.1↑) |
| ResNet18 with Faster R-CNN | | | | |
| HAQ | 3MP/3MP | 22.3 | 82 | 69.0 |
| HAQ+A | 3MP/3MP | 21.8(0.5↓) | 76(5↓) | 69.3(0.3↑) |
| HAWQ | 3MP/3MP | 26.3 | 85 | 66.4 |
| H+A | 3MP/3MP | 25.9(0.4↓) | 81(4↓) | 66.6(0.2↑) |

than SOTA methods on VOC. Moreover, directly using the policy found by HAQ and EdMIPS on CIFAR10 to quantize target model suffers from significant performance degradation on VOC. Since the mixed-precision models needs to be pretrained on ImageNet, the search cost decrease on VOC is more sizable than that on ImageNet.

**ResNet.** Table 2 also shows the results of ResNet18 with Faster R-CNN in object detection task. After applying ASGA, EdMIPS can achieve the same accuracy as before with only 77% of the search cost, which significantly improves the search efficiency.

### 3.3. Ablation Study

#### 3.3.1. EFFECTIVENESS OF $\rho$

In fact, sharpness-awareness method is sensitive to the perturbation radius $\rho$. Especially, an excessively large $\rho$ would incur a sharp decline in generalization performance, while an excessively small one may cause the model to learn the noise or high-frequency details in the images instead of generalized features. Figure 4 shows the comparison of MPQ policy search accuracy and the convergence epochs of ResNet18, ResNet50, and MobileNet-V2 with different fixed and adaptive $\rho$ on CIFAR10, respectively. We observe that adaptivity $\rho$ achieves faster convergence compared to a fixed $\rho$ without accuracy degradation.

Table 3. Comparison of MPQ policies search results for ResNet18 on different datasets. $\sigma_{MA}$ and $\sigma_{MA}$-A represent the average $\sigma_{max}$ for the baselines without/with ASGA, respectively.

| Datasets | $\sigma_{MA}$ | $\sigma_{MA}$-A | Top-1 | Top-5 |
|---|---|---|---|---|
| CIFAR10 | 13.5 | 11.4 (2.1↓) | 90.86 | 99.62 |
| Flowers | 48.4 | 44.4 (4.0↓) | 87.32 | 98.75 |
| Food | 25.8 | 23.1 (2.7↓) | 84.60 | - |

#### 3.3.2. EFFECTIVENESS OF PROXY DATASETS

There is no relevant literature to study the effect of proxy subsets. The searched results of our method on different small-scale proxy datasets, including CIFAR10, Flowers and Food, are shown in Table 3, where for the baseline, we set $\rho_0 = 0.1$ for calculating sharpness. Compared to the baseline without sharpness regularization, the one with ASGA achieves flatter loss landscape across all datasets, and gets the best accuracy on CIFAR10. As shown in Figure 6, using CIFAR10 as proxy dataset, we can find more well-performing MPQ policies than other datasets. This may be because the categories of CIFAR10 are more similar to those of target ImageNet compared to other datasets. In this sense, a proxy dataset with more class-level similarity to target dataset could be considered to improve the performance better. Figure 5 shows the weight and activation bitwidth assignment for ResNet18, ResNet50 and MobileNet-V2.

#### 3.3.3. EFFECTIVENESS OF TARGET DATASET'S SUBSET

We randomly select 60K images from target dataset ImageNet (i.e., 0.45% of target ImageNet training data) to form a subset, whose size is similar to that of CIFAR10. We use them to search for a 3MP policy for ResNet18, and the results are shown in Table 4. From this table, we obtain following observations: (1) the subset of ImageNet without proposed ASGA suffers about 1.6% performance degradation compared to CIFAR10 with ASGA. This demonstrates that the data distribution in the subset is still different from the full set, which necessitates the adoption of our ASGA. (2) For both subset and CIFAR10, the use of our ASGA leads to obvious accuracy improvement over the baseline. These results again validate the effectiveness of our ASGA method.

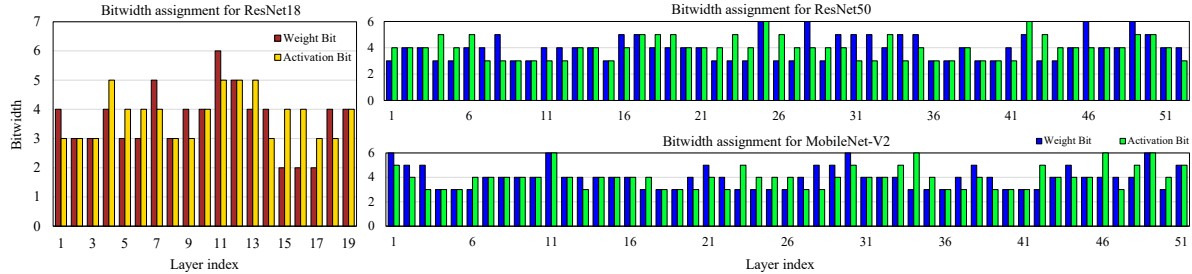

Figure 5. Bitwidth assignment for each layer of ResNet18, ResNet50, and MobileNet-V2

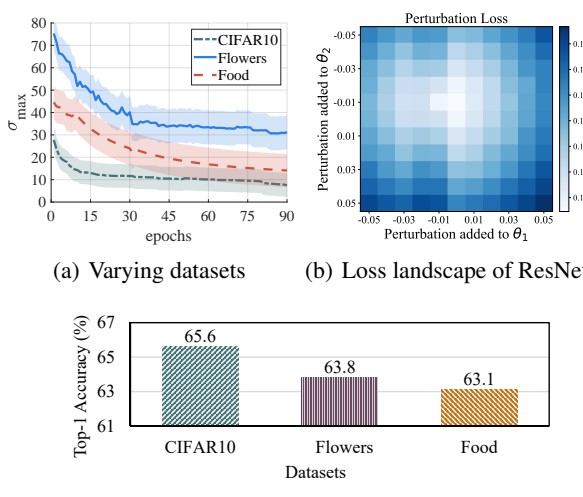

(a) Varying datasets    (b) Loss landscape of ResNet18

(c) Top-1 accuracy on ImageNet

Figure 6. (a) Comparison of $\sigma_{max}$ variations for ResNet18 on three datasets. (b) The heatmap of partial loss landscape of ResNet18 on CIFAR10. White denotes a lower perturbed loss and blue denotes a higher one. (c) Comparison of Top-1 accuracy on ImageNet for MPQ policy searched on three proxy datasets. The MPQ on CIFAR10 obtains the lowest $\sigma_{max}$ and highest Top-1 on ImageNet.

## 4. Related Work

**Mixed-Precision Quantization.** Unlike FPQ (Zhou et al., 2016) assigning uniform bitwidth for all layers, MPQ aims to allocate different bitwidths to different layers (Elthakeb et al., 2020). The key challenge to realize this is how to find the optimal bitwidth for each layer from the exponential discrete bitwidth space. A series of intelligent methods have been developed for solving the above issue, e.g., HAQ (Wang et al., 2019), SPOS (Guo et al., 2020), and EdMIPS (Cai & Vasconcelos, 2020). Especially, DNAS (Wu et al., 2018) and BP-NAS (Yu et al., 2020) formulate the MPQ process as a Neural Architecture Search (NAS) problem to search for optimal MPQ policy. Also, there exist several once quantization-aware methods, e.g., HAWQ (Dong et al., 2019) (Yao et al., 2021) and MPQCO (Chen et al., 2021), which exploit the second-order quantization information to learn optimal bitwidths. Recently, GMPQ (Wang et al., 2021) proposes to seek optimal policy by using attribution

Table 4. Results of the ablation study on the subset of ImageNet. Subset refers to the subset extracted from ImageNet. "Top-1/5" represents Top1/5 accuracy respectively. ✓ and ✗ represent the baseline method (EdMIPS) with/without ASGA, respectively.

| Datasets | ASGA | Top-1 | Top-5 |
|----------|------|-------|-------|
| Subset | ✗ | 66.3 | 85.6 |
| Subset | ✓ | 67.3 (1.0↑) | 86.9 (1.3↑) |
| CIFAR10 | ✗ | 66.4 | 86.1 |
| CIFAR10 | ✓ | 67.9 (1.5↑) | 87.7 (1.6↑) |

rank consistency, and SEAM (Tang et al., 2023) introduces the class margin regularization to enhance MPQ search capability. However, both of them suffer from intricate calculation of feature maps, increasing search costs.

**Sharpness and Generalization.** Recent studies (Keskar et al., 2016; Cha et al., 2021; 2022) have revealed direct relationship between sharpness and generalization capability of DNNs. Keskar et al. (Keskar et al., 2016) make the first attempt to offer a sharpness measure and demonstrate its correlation with the generalization ability. Dinh et al. (Dinh et al., 2017) further argue that the sharpness measure bears a relation with the spectrum of the Hessian. SWAD (Cha et al., 2021) provides a theoretical insight that a flatter minimum can lead to smaller generalization gap under some out-of-distribution conditions. Then, SAM (Foret et al., 2020) aims to identify parameter regions that exhibit relative "flatness" in the vicinity of the low loss landscape, leading to a set of SAM improvements, e.g., ASAM (Kwon et al., 2021), Fisher SAM (Kim et al., 2022), and GSAM (Zhuang et al.). Several studies (Dung et al., 2024) have utilized sharpness to enhance data generalization for quantization. However, no existing works attempt to apply the concept of sharpness for realizing generalization of MPQ.

## 5. Conclusion

In this work, we propose ASGA-based DMPQ to search for transferable quantization policy with small amount of data. To bridge the generalization gap, we not only center around enhancing the accuracy on small dataset, but also minimizing the sharpness measure of the loss landscape in MPQ search. We consider this as a loss-sharpness information exploitation on small proxy dataset, which shows a

dataset-independent attribute. Then, we design an adaptive sharpness-aware gradient aligning (ASGA) method and introduce it into differentiable MPQ search paradigm, aiming to acquire transferable MPQ policy from small dataset to target large dataset. The theoretical analysis and experimental results validate our idea, and we use only small dataset with 0.5% the size of large one to search for quantization policy, achieving equivalent accuracy on large datasets with speeding up the search efficiency by up to 150%.

## Acknowledgements

This work is supported in part by the National Natural Science Foundation of China under Grant 62472079 and the Fundamental Research Funds for the Central Universities (No.N2417003).

## Impact Statement

Mixed Precision Quantization (MPQ) becomes a promising technique for compressing deep model, but suffers from high search burden. This paper solves this issue via introducing a generalizable quantization paradigm, motivated by learning from loss landscape. Our approach provides fresh insights for advancing MPQ. This work has no ethical aspects as well as negative social consequences.

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

## Overview of the Supplementary Material

The main contents of this Supplementary Material are as follows:

## A. Supplementary Material

### A.1. Differences between Our Method and Conventional MPQ Methods

Figure 7 demonstrates the difference between our proposed method and conventional MPQ mehohds. More specifically, our proposed ASGA-based GMPQ method (termed as GMPQ-ASGA) aims to learn transferable MPQ policy via loss sharpness optimization for efficient inference. Unlike existing methods that require the dataset consistency between quantization policy search and model deployment, our approach enables the acquired MPQ policy to be generalizable from proxy datasets to target large-scale datasets. The MPQ policy found on small-scale datasets achieves promising performance on challenging large-scale datasets, so that the MPQ search burden is largely alleviated.

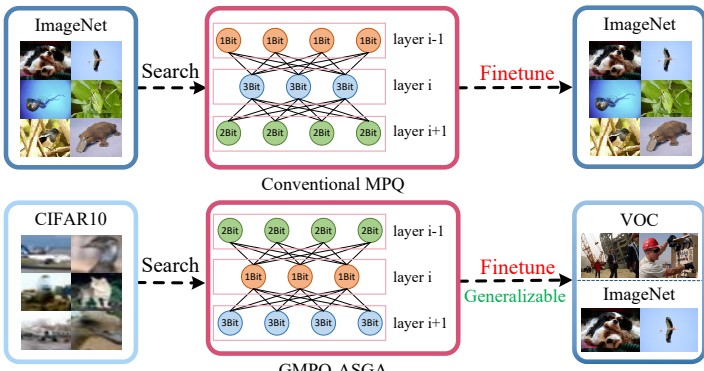

Figure 7. Conventional methods require the consistency of datasets for bitwidth search and model deployment, while our GMPQ-ASGA searches the optimal quantization policy on small datasets and generalizes it to large-scale datasets.

### A.2. Details of Adaptive Sharpness-Aware Gradient Aligning

A.2.1. DETAILS OF THE DIRECTIONAL CONSISTENCY AMONG $\nabla h(\theta)$, $\nabla \mathcal{L}(\theta)$, AND $\nabla \mathcal{L}_p(\theta)$

In Section 2.2, we know that:

$$\nabla h(\theta) = \nabla \mathcal{L}_p(\theta) - \nabla \mathcal{L}(\theta). \tag{20}$$

According to Eq. (20), when the directions of $\nabla \mathcal{L}(\theta)$ and $\nabla \mathcal{L}_p(\theta)$ are the same, there exists a scalar $\eta$ such that:

$$\nabla h(\theta) = \nabla \mathcal{L}_p(\theta) - \nabla \mathcal{L}(\theta) = \eta \nabla \mathcal{L}(\theta) - \nabla \mathcal{L}(\theta) = (\eta - 1)\nabla \mathcal{L}(\theta). \tag{21}$$

When $\eta > 1$, the direction of $\nabla h(\theta)$ is exactly the same as that of $\nabla \mathcal{L}(\theta)$, and its length is expanded by $(\eta - 1)$. When $0 < \eta < 1$, $\nabla h(\theta)$ and $\nabla \mathcal{L}(\theta)$ are still in the same direction, but the length of $\nabla h(\theta)$ is reduced. This means that, regardless of the value of $\eta$, $\nabla h(\theta)$ always maintains the same direction as $\nabla \mathcal{L}(\theta)$. Therefore, we need to align the gradients of $\mathcal{L}(\theta)$ and $\mathcal{L}_p(\theta)$ so that $h(\theta)$ can be effectively optimized under the above-mentioned circumstances.

A.2.2. DETAILS OF GRADIENT ALIGNMENT

In this section, we present the detailed design of the gradient alignment in ASGA. Following the work (Wang et al., 2023), we perform Taylor expansion of $\mathcal{L}_p(\theta)$ on Eq. (9) and obtain:

$$
\begin{aligned}
&\min_\theta(\mathcal{L}_p(\theta - \mu\nabla\mathcal{L}(\theta))) \\
&= \min_\theta(\mathcal{L}_p(\theta) - \mu\nabla\mathcal{L}_p(\theta) \cdot \nabla\mathcal{L}(\theta) + o(\mu\nabla\mathcal{L}(\theta))) \\
&\approx \min_\theta(\mathcal{L}_p(\theta) - \mu\nabla\mathcal{L}_p(\theta) \cdot \nabla\mathcal{L}(\theta)),
\end{aligned}
\tag{22}
$$

where $o(\mu\nabla\mathcal{L}(\theta))$ represents a higher-order infinitesimal term that becomes negligible as $\mu$ approaches 0. Intuitively, we can see that minimizing Eq. (22) is actually equivalent to minimizing $\mathcal{L}_p(\theta)$ while maximizing the inner product of $\nabla\mathcal{L}_p(\theta)$ and $\nabla\mathcal{L}(\theta)$.

More specifically, minimizing $\mathcal{L}_p(\theta)$ is conductive to searching for a sufficiently low minimum, and minimizing the inner product can ensure that the loss landscape near the minimum is sufficiently flat. If the gradient direction of $\nabla\mathcal{L}_p(\theta)$ is substantially similar to that of $\nabla\mathcal{L}(\theta)$, their inner product will be greater than 0, and achieves the maximum value when the directions are completely aligned.

As stated above, we can clearly see that the gradient direction of $h(\theta)$ is influenced by both $\nabla\mathcal{L}(\theta)$ and $\nabla\mathcal{L}_p(\theta)$. If $\nabla\mathcal{L}(\theta)$ and $\nabla\mathcal{L}_p(\theta)$ deviate significantly, severe gradient conflict will prevent the simultaneous optimization of all the three loss objectives. Conversely, when $\nabla\mathcal{L}(\theta)$ and $\nabla\mathcal{L}_p(\theta)$ are aligned, the three will be optimized efficiently and consistently. In this sense, aligning $\nabla\mathcal{L}(\theta)$ and $\nabla\mathcal{L}_p(\theta)$ facilitates the achievement of the optimization objectives in the given formula.

**A.3. Details of adaptive strategy of controlling $\rho$**

According to Section 2.2, we know that $\rho$ should be gradually increased to effectively continue optimizing the sharpness of the model. Therefore, the value of $\rho$ should meet the following requirements:

- It must always be greater than 0;

- There must be an upper limit $\rho_{max}$;

- It should adaptively increase as the $h(\theta)$'s increase rate decreases.

Based on the aforementioned considerations, we formulate an adaptive strategy to adjust the value of $h(\theta)$ according to the status of sharpness of the loss landscape, which is defined as

$$
\rho = \min(\rho_{max}, \frac{\phi}{\ln(h(\theta) + 1)}).
\tag{23}
$$

Our design has the following merits:

(1) The characteristics of the logarithmic function in Eq. (23) ensure that when $h(\theta) > 0$ (which is also the case in reality), $\rho$ is greater than 0.

(2) The term $\rho_{max}$ in Eq. (23) ensures that $\rho$ will not increase indefinitely, thus preventing instability issues in the optimization process.

(3) In the early stages of policy search, the value of $h(\theta)$ is relatively large, while the value of $\frac{\phi}{\ln(h(\theta)+1)}$ is small. As the loss landscape becomes increasingly flatter, the value of $h(\theta)$ is gradually decreasing, and $\frac{\phi}{\ln(h(\theta)+1)}$ is increasing accordingly. This dynamic characteristic allows the value of $\rho$ to be adaptively adjusted according to the changes in $h(\theta)$, avoiding the issue of a fixed $\rho$ being unable to sensitively capture changes when $h(\theta)$ is small.

Moreover, the scaling factor $\phi$ in Eq. (23) is able to freely adjust the value range of $\rho$ to adapt to different tasks, as shown in Figure 8, which illustrates the curves of $\rho$ varying with $h(\theta)$ under different combinations of $\rho_{max}$ and $\phi$.

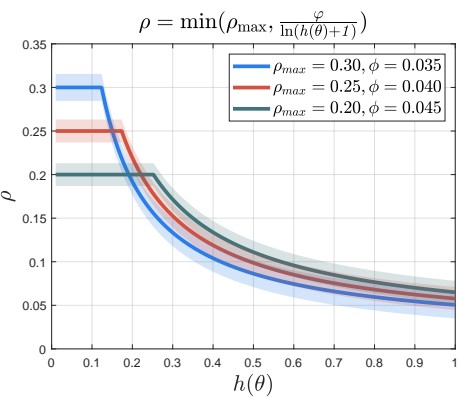

Figure 8. The curves of $\rho$ varying with $h(\theta)$ under different combinations of $\rho_{max}$ and $\phi$.

## A.4. Proof of Lamma 1.

**Lemma 1.** *Suppose the training set contains $m$ elements drawn i.i.d. from the true distribution and the average loss in MPQ search is $\mathcal{L}(\theta) = \frac{1}{m}\sum_{i=1}^{m}\mathcal{L}_i(\theta, x_i, y_i)$ ($\mathcal{L}_i(\theta)$ for short), where $(x_i, y_i)$ is the input-target pair of $i$-th element. Suppose $\theta$ is drawn from posterior distribution $\tau$. For the prior distribution (independent of training) $\zeta$, we have:*

$$\mathbb{E}_{\theta\sim\tau}\mathbb{E}_{x_i}\mathcal{L}_i(\theta) \leq \mathcal{L}_p(\theta) + \mathcal{R}, \tag{24}$$

*with probability at least $(1 - a)\left[1 - e^{-\left(\frac{\rho}{\sqrt{2}b} - \sqrt{k}\right)^2}\right]$, where $a \in (0, 1)$ denotes confidence level, and $\mathcal{R} = 4\sqrt{\left(KL(\tau\|\zeta) + \log\frac{2m}{a}\right)/m}$. This shows that ASGA effectively reduces the upper bound of the generalization error during the MPQ process.*

**Proof.** According to the PAC-Bayeian theory (McAllester, 2003), we can employ Probably Approximately Correct (PAC) inequalities to establish the generalization bounds for various machine learning models. Here, the generalization bound reefers to the upper limit on the divergence between the model's empirical loss (estimated from training data) and the population loss (which is the loss incurred on the true, underlying data distribution). The optimization of these bounds fosters the emergence of self-certified learning algorithms. According to the Bayesian theory, for a well-trained quantized model derived from the optimal MPQ policy, the upper bound of its expected loss at convergence can be expressed as:

$$\mathbb{E}_{\theta\sim\tau}\mathbb{E}_{x_i}\mathcal{L}_i(\theta) \leq \mathbb{E}_{\theta\sim\tau}\mathcal{L}(\theta) + \mathcal{R}. \tag{25}$$

Here, $\mathcal{L}(\theta)$ can be considered as a local minimum at this point, and then $\mathcal{L}_p(\theta)$, which is defined in Eq. (6), is always greater than $\mathcal{L}(\theta)$. As a result, the following inequality holds:

$$\mathbb{E}_{\theta\sim\tau}\mathbb{E}_{x_i}\mathcal{L}_i(\theta) \leq h(\theta) + \mathcal{L}(\theta) + \mathcal{R}, \tag{26}$$

with a probability of $(1 - a)\left[1 - e^{-\left(\frac{\rho}{\sqrt{2}b} - \sqrt{k}\right)^2}\right]$ less than $(1 - a)$, which implies that minimizing $h(\theta)$ is expected to achieve a tighter upper bound of the generalization performance. Therefore, Lemma 1 is proven. Moreover, $\mathcal{R}$ in Eq. 26 is typically hard to analyze and often simplified to L2 regularization. Note that $\mathcal{L}_p(\theta) = h(\theta) + \mathcal{L}(\theta)$ only holds when $\rho$ equals $\rho_{ture}$ (the ground truth value determined by underlying data distribution); when $\rho \neq \rho_{ture}$, $\min(\mathcal{L}_p(\theta), h(\theta))$ is more effective than $\min\mathcal{L}_p(\theta)$ in terms of minimizing generalization loss.

## A.5. Datasets and Implementation Details

**Details of datasets.** CIFAR10 consists of 60K images of 32 X 32, divided into 10 categories. Flowers has 102 categories and each category involves 40 to 258 images. Food contains 32135 high-resolution physical images from 6 restaurants. For image classification, the target large dataset $\mathcal{D}_{val}$ for model inference is ImageNet (Deng et al., 2009) with 1000 categories, containing 1.28M training samples and 50K validation samples. For object detection, the target dataset is VOC (Everingham et al., 2010) with 20 categories, containing about 1.6K training samples and 5K validation samples.

**Details of models.** We employ four representative network architectures for experiments, including: (1) ResNet-18, 50 (He et al., 2016), MobileNet-V2 (Sandler et al., 2018) for image classification, and (2) ResNet-18 and VGG16 (Simonyan & Zisserman, 2014) for object detection.

**Details of parameters.** For policy searching, we adopt the SGD as base optimizer, and the initial learning rate is set to 0.01 for 90 epochs. Empirically, we find the sharpness of loss landscapes is not sensitive to the hyper-parameter and thus set $\epsilon$ = 0.1 for all proxy datasets. We use the full-precision model (trained on $\mathcal{D}_{train}$) as the initialization and adopt the SDG optimizer with Nesterov momentum (Sutskever et al., 2013) and the initial learning rate is set to 0.04. We use the cosine learning rate scheduler and finetune the model until convergence and the first 5 finetune-epochs are used as warm-up.

The key notations and experimental parameters used in this paper are listed in Table 5 and Table 6, respectively.

Table 5. Key notations in this paper.

| Notation | Description |
|---|---|
| $\mathcal{L}$ | the loss value of the model |
| $\mathcal{L}_p$ | the perturbation loss value of the model |
| $\mathcal{L}_{val}$ | the task loss on the validation dataset |
| $\mathcal{L}_{train}$ | the task loss on the train dataset |
| $\mathcal{L}_{comp}$ | the complexity loss |
| $\theta$ | the weights set of quantized network |
| $\theta^*$ | the optimal weights set of quantized network |
| $\mathcal{D}_{train-proxy}$ | the proxy dataset in MPQ search stage |
| $\mathcal{D}_{val}$ | the dataset containing all validation data in model inference stage |
| $\mathcal{D}_{train}$ | the dataset containing all training data in model inference stage |
| $\mathcal{Q}$ | the quantization policy |
| $\Omega$ | the resource-constrained conditions |
| $\delta$ | the perturbations to model weights |
| $\rho$ | the perturbations radius |
| $\phi$ | the scaling factor to control $\rho$ |
| $h$ | the surrogate gap |
| $\sigma_{max}$ | the upper limit of the value of $\rho$ |
| $\mu$ | the scaling factor to control the ascent step |
| $\epsilon$ | the scaling factor for weighting the corresponding loss in the optimization process |
| $\tau$ | the posterior distribution of $\theta$ under the optimal MPQ policy |
| $\zeta$ | the prior distribution of $\theta$ |
| $a$ | the confidence level |
| $B^\theta$ | the predefined bitwidth candidate sets for weights |
| $B^a$ | the predefined bitwidth candidate sets for activations |

Table 6. Key experimental parameters in the paper.

| Parameter | Scope of application | Setting |
|---|---|---|
| epochs for searching | all | 90 |
| base optimizer | all | SGD |
| learning rate | searching stage | 0.01 |
| | inference stage | 0.04 |
| $\epsilon$ | all | 0.1 |
| $\rho_0$ | all | 0.1 |
| $B^\theta$ | MobileNet-V2, VGG16 | $\{2,3,4,5,6\}$ |
| | ResNet18, ResNet50 | $\{2,3,4,6\}$ |
| $B^a$ | MobileNet-V2, VGG16 | $\{2,3,4,5,6\}$ |
| | ResNet18, ResNet50 | $\{2,3,4,6\}$ |

