# OpenReview forum: "Learning from Loss Landscape: Generalizable Mixed-Precision Quantization via Adaptive Sharpness-Aware Gradient Aligning"
_ICML.cc/2025/Conference — ICML 2025 poster_

### Official Review · Reviewer_WvMG · 2025-03-09

**Overall Recommendation:** 3

**Summary:**

This paper propose a search-based method to find mixed-precision quantization scheme that can work with a much smaller dataset. To enable the generalization from the small search set to the large validation set, the paper propose to seek for quantization scheme that can lead to flatter loss minima of the quantized model. This leads to the proposal of using an adaptive sharpness-aware gradient aligning objective in the search process.

## update after rebuttal

Post rebuttal, I agree with the author that the proposed method can help on calibrating the model with different datasets, no matter which policy search method is used. To this end, I increase my score to weak accept.

**Claims And Evidence:**

The claim that using sharpness information can help the search of a more generalizable quantization scheme is supported by both theortical analysis and experiment results. However, the claimed advantage of the proposed method is not justified. For example:
1. The paper claims that searching MPQ policy with subset would reduce performance, yer this is only a drawback of differentiable search-based methods, not for other more advanced mixed precision quantization methods like HAWQ. This claim is inaccurate and hinders the motivation of the proposed method.
2. In related work, the paper mentions that previous method suffer from intricate calculation of feature maps, increasing search costs. However, it seems apparent that the gradient computation required for the proposed ASAM is more costly than the feature map computation. The improved seearch cost claimed by the proposed method is only supported by the reduced numbre of epochs, yet ignoring the potential higher cost in each epoch.

**Essential References Not Discussed:**

References are adequate.

**Experimental Designs Or Analyses:**

As reported in Table 1, HAWQ appears to be a significantly stronger baseline than other DNAS-based methods, yet HAWQ is not used as baseline in other settings. I believe HAWQ shoudl be compared with in all settings to prove the contribution of the proposed method is effective. Moreover, HAWQ performs a single-shot ILP-based search followed by finetuning. It is unclear how the H+A is performed in Table 1 and what con-epoch stands for here.

**Methods And Evaluation Criteria:**

The evaluation is conducted cross multiple models and datasets. The setting is standard for quantization literature. However, the paper uses "con-epochs" as a key criteria in the experiments, which is not a fair criteria. The proposed method relies on SAM, which requires at least two back propagations per epoch, effectively increasing the convergence time over baseline methods even with a reduced epoch counts. Real training time should be reported here as a better criteria.

**Other Comments Or Suggestions:**

Post rebuttal, I agree with the author that the proposed method can help on calibrating the model with different datasets, no matter which policy search method is used. To this end, I increase my score.

**Other Strengths And Weaknesses:**

Besides the weakness mentioned prevously on the motivation, baseline, and evaluation metrics, SAM-based optimization has a fundamental drawback of complexity, which makes the convergence challenging on more complicated architectures like transformers. The exploreation of the proposed method is limited to simple CNN models in this paper. The scalability of the search and the SAM objective to transformer models is questionable.

**Questions For Authors:**

Please address the weaknesses mentioned in previous parts.

**Relation To Broader Scientific Literature:**

This paper aims to improve over previous DNAS-based mixed precision quantization method by including SAM-based objective in the search process. As both DNAS-based methods and SAM objectives are well studied, this specific usage appears to be novel. However, it shoudl be noted that DNAS-based methods are not promising compred with ILP-based methods like HAWQ in inducing mixed-preciison quantization scheme. This fundamentally hinders the significance of the proposed method.

**Theoretical Claims:**

I checked the method derivation from Equ.(1) to (12). However, I did not check the theortical analysis in Sec. 2.4.

---

> ### Author Rebuttal · Authors · 2025-03-31
>
> ## Q1 The claims of ASGA advantages and search cost.
>
> A1: Many thanks. We think the ambiguity of claim's presentation may cause your confusion of this work. Our experiments (**Table 1 in Section 3 of our paper**) show that **using ASGA with ResNet50 saves 16 GPU hours with better Top-1 compared to HAWQ alone, which validates our motivation**. We explain this as below. HAWQ's second-order information of the Hessian matrix is data sensitive[1]. It works well if the proxy subset distribution matches target domain; otherwise, performance degradation occurs since distribution shift may disrupt Hessian estimation. In fact, proxy dataset is often distribution-discrepant, especially for privacy-sensitive scenarios where directly getting subset is not allowed. Conversely, ASGA’s generalizable policy—searched on proxy dataset—can obtain satisfactory performance with large reduction of search time on target datasets (**Table 1**). We will improve related presentations (related work).  As for search cost, we have added comparison experiments with GPU hours evaluation (**see A2**).
>
> ---
>
> [1]: Hawq: Hessian aware quantization of neural networks with mixed-precision, ICCV 2019.
>
> ---
>
> ## Q2 Is the evaluation criterion fair?
>
> A2: We have added experimental results of real training time as below (GH means GPU hours). However, we still believe con-epoch can be used for our scenarios. Rethink our two-step workflow: “X+A” means that ASGA first finds generalizable MPQ policy with small proxy data, and then baseline X finetunes this policy on target dataset with several epochs, where ASGA is not involved, and thus the search cost of each baseline at each epoch is **not increased significantly by ASGA**. Note that, the generalizable policy of model found by ASGA can be suitable to the baselines that require the same model structure, which **largely reduces search burden**. Without ASGA, baselines should train from scratch on large dataset, requiring much search time (**Table 1**).
>
> |Model| Methods | W/A | con-epoch |Top-1 | GH of ASGA | GH/epoch | Total GH  |
> | :------: | :-----: | :---------: | :-------: | :-----: | :--------: | :------: | :-------: |
> | ResNet50 |  HAWQ   |   4MP/4MP   |    98     |   74.6   |     -     |   3.04   |   298.2   |
> |          | **H+A** | **4MP/4MP** |  **92**   | **74.9** |  **2.4**   | **3.05** | **283.1** |
> | ResNet18 |  GMPQ   |   3MP/3MP   |    96     |   66.3   |   -      |   1.24   |  119.05   |
> |          | **G+A** | **3MP/3MP** |  **80**   | **66.4** |  **2.1**   | **1.24** | **101.5** |
> | ResNet18 |  SEAM   |   3MP/3MP   |    96   |   65.1   | - |   1.11   |  106.53   |
> |          | **S+A** | **3MP/3MP** |  **85**  | **65.8** | **2.1** | **1.11** | **96.48** |
> ## Q3 Compare with HAWQ as baseline in other settings.
>
> A3: We need to clarify that "H+A" refers to using HAWQ to finetune the MPQ policy searched by ASGA, as presented in **A2**, while "con-epoch" denotes the number of iterations required for convergence. In this sense, both DNAS and HAWQ can be used as baseline to utilize the policy found by ASGA. Then, we have added experimental results under ResNet18 model, which show that by fine-tuning the policy of ASGA, HAWQ **saves 2.7 GH** (GH means GPU hours). Due to time limitation, the experiments on MobileNet-V2 will be provided in the future.
>
> |  Model  | Methods |   W/A   | con-epoch |  Top-1   | GH of ASGA | GH/epoch | Total GH  |
> | :-----: | :-----: | :---------: | :-------: | :------: | :--------: | :------: | :-------: |
> | Reset18 |  HAWQ   |   4MP/4MP   |  95   |   67.3   |     -      |   1.14   |   108.3   |
> |         | **H+A** | **4MP/4MP** |  **90**   | **67.4** |  **2.1**   | **1.15** | **105.6** |
>
>
> ## Q4 The significance of ASGA.
>
> A4: Notice that both DNAS- and ILP-based MPQs can employ ASGA for reducing search cost without sacrificing performance, which demonstrate the significance of the proposed method. In fact, our ASGA is **suitable to the enhancement of ILP-based MPQs**, for instance, HAWQ-V3 can quickly finetune ASGA’s policies for stable performance across different datasets and hardware. Besides of DNAS-based methods, **ASGA offers a flexible and efficient solution for enhancing various MPQs**, especially for resource-limited, cross-dataset scenarios.
>
> ## Q5 The scalability of ASGA on Transformer.
>
> A5: **ASGA is applicable to various models including Transformer**, since its core design—minimizing loss landscape sharpness — is not CNN-specific. In fact, [2] have demonstrated the scalability of SAM, which is applied to Transformers (BERT and ViT). Note that current MPQ researches (including SOTA works like EdMIPS and HAQ) primarily focus on CNNs, and thus our experiments also central on CNN models rather than Transformers. We will extend our work to Transformers in future research—a direction that is theoretically justified and critical for advancing practical MPQ deployments.
>
> ---
>
> [2]: Towards efficient and scalable sharpness-aware minimization, CVPR 2022.
>
> ---

---

### Official Review · Reviewer_NNU4 · 2025-03-10

**Overall Recommendation:** 3

**Summary:**

This paper aims to reduce the mixed-precision quantization search costs by decoupling the policy search and model deployment dataset. In this way, the mixed-precision quantization policy of a model can be searched on a small-scale dataset and then the policy can be transferred to a large-scale one for deployment. To this end, the authors introduce sharpness-aware minimization (SAM) to ensure generalization, as well as a gradient alignment method to handle gradient conflicts and an adaptive perturbation radius to accelerate optimization.

**Claims And Evidence:**

Yes.

**Essential References Not Discussed:**

While this submission provides basic discussion for mixed-precision quantization, it lacks of published papers in recent 3 years, e.g., [1][2][3][4]. Moreover, baselines are really old, e.g., EdMIPS (CVPR'20), GMPQ (ICCV'21), HAQ (CVPR'19).

[1] Retraining-free Model Quantization via One-Shot Weight-Coupling Learning, CVPR 2024

[2] One-shot Model for Mixed-Precision Quantization, CVPR 2023

[3] SDQ: Stochastic Differentiable Quantization with Mixed Precision, ICML 2022

[4] Mixed-Precision Network Quantization via Learned Layer-wise Importance, ECCV 2022

**Experimental Designs Or Analyses:**

I think the authors could provide more comparisons about recent published papers.

**Methods And Evaluation Criteria:**

I appreciate the authors' efforts, however, [1] already explored the feasibility of combining SAM with quantization. While I see the authors aim to organize this submission from another perspective (i.e., decoupling the search/deployment datasets), the perspective itself is still not new. Therefore, this submission looks like a bit A+B for me now.

Moreover, I think at least some discussion and primary experiments should be provided in this submission, to further distinguish differences between the proposed method and [1].


[1] Sharpness-aware Quantization for Deep Neural Networks

**Other Comments Or Suggestions:**

N/A

**Other Strengths And Weaknesses:**

I checked the two most related works, GMPQ and SEAM, and found the Fig.2, and Equ (4)(5) of this paper are quite similar to SEAM, and Fig.7 is quite similar to GMPQ.

**Questions For Authors:**

Combining quantization with SAM seems to be promising but leads to an additional question, can they improve the generalization performance? In other words, does the searched mixed-precision quantization policy provide better generalization performance compared to existing works (SEAM, GMPQ)?

**Relation To Broader Scientific Literature:**

This work extends the previous works (GMPQ, SEAM).

**Theoretical Claims:**

Checked.

---

> ### Author Rebuttal · Authors · 2025-03-31
>
> ## Q1 The novelty of the work.
>
> A1: Many thanks for your comment! This is a valuable question. The method SAQ [1] you mentioned appears similar to our work, but **they are actually different**. First, the goal of our work is **fundamentally different** from such work. SAQ quantizes and trains target model **on the same dataset**, aiming to enhance model's anti-interference ability by smoothing the loss function to prevent the occurrence of overfitting. In contrast, our ASGA aims to significantly boost the search efficiency on large dataset via searching for a generalizable MPQ policy on a small-scale proxy dataset, enabling it to achieve SOTA accuracy with fewer computational resources on a large-scale target dataset (**Table 1 in Section 3 of our paper**). The advantage of ASGA lies in the improvement of **cross-dataset generalization ability**. Second, SAQ focuses on **non-differentiable fixed-precision quantization**, particularly uniform quantization scenarios, whereas ASGA is designed for **mixed-precision quantization**. Comparative studies[2][3] demonstrate that MPQ **achieves better complexity-accuracy** balance in practical applications. Moreover, we emphasize that **ASGA is not merely an "A+B" combination**. Beyond separating search and deployment datasets, ASGA's primary contribution lies in optimizing the existing MPQ training paradigm, **offering novel solutions for cross-dataset generalization, convergence efficiency, and computational resource utilization**.
>
> ---
>
> [1]: Sharpness-aware quantization for deep neural networks[J]. arXiv preprint arXiv:2111.12273, 2021.
>
> [2]: Haq: Hardware-aware automated quantization with mixed precision, CVPR 2019.
>
> [3]: Towards mixed-precision quantization of neural networks via constrained optimization, ICCV 2021.
>
> ---
>
> ## Q2 Literature supplementation and old baselines.
>
> A2: Thanks for the advice. We'll cite these papers in revision. Regarding outdated baselines, according to the available source code in the papers you provided, we add comparative experiments on ResNet18 (GH means GPU hours). The results show that with ASGA-searched MPQ, RFQuant obtains up to **1.11X searching efficiency improvement**, further validating our method's effectiveness.  Due to time limitation, the comparison experiments with more new baselines will be conducted in our subsequent research.
>
> |  Methods|   W/A   | con-epoch |  Top-1   | Top-5 | GH of ASGA | GH/epoch | Total GH  |
> | :-----------: | :---------: | :-------: | :------: | :------: | :--------: | :------: | :-------: |
> |  RFQuant[4]  |   4MP/4MP   |94 | 65.8 | 84.4 | - |1.30| 122.4 |
> | **R[4] + A** | **4MP/4MP** |**85** | **65.9** | **84.9** |  **2.1**   | **1.31** | **113.4** |
>
> ---
>
> [4]: Retraining-free model quantization via one-shot weight-coupling learning, CVPR 2024.
>
> ---
>
> ## Q3 The differences between ASGA, SEAM and GMPQ.
>
> A3: Thanks for your comment! In fact, both ASGA and GMPQ have similar goal, i.e., find generalizable policy across datasets, and thus share similar quantization process like Fig.7. Regarding Figure. 2 and Eqs.4-5, both SEAM and ASGA build upon the differentiable MPQ framework, and thus share several differentiable search formulations. However, our method is **fundamentally distinct from SEAM and GMPQ**: SEAM seeks generalizable MPQ policy by minimizing intra-class compactness and maximizing inter-class separation, and GMPQ exploits attribution consistency and it relies on a pre-trained full-precision model for quantized model alignment, requiring extra training costs; Different from them, ASGA focuses on loss sharpness optimization with **no need pre-trained model**, which is more efficient.  For sake of clarity, we will modify the related figures and equations located by reviewers in the revision.
>
> ## Q4 Performance improvement of ASGA compared with GMPQ and SEAM.
>
> A4: Many thanks. First, many studies[5][6] have theoretically proven that minimizing loss landscape sharpness can well enhance model’s generalization ability. Then, we conducted extensive experiments to validate the superiority of ASGA over SEAM and GMPQ. Experimental results in **Table 1** show  the quantized model derived from MPQ policy found by ASGA (for ResNet18) **achieves a Top-1 improvement of 0.7% and 0.1% on ImageNet over those of SEAM and GMPQ**, respectively, and **save training costs by 17% and 11.5%**, respectively. Moreover, the comparative experiments on a set of baselines in **Table 1** show that by using a small dataset of **only 0.5% the size of the large dataset** to search for quantization policies, ASGA **achieves the same accuracy on the large dataset and improves the search efficiency by up to 150%**. These validate **the superiority of ASGA in improving  generalization of quantization policies over both SEAM and GMPQ**.
>
> ---
>
> [5]: Fisher sam: Information geometry and sharpness aware minimisation, ICML 2022.
>
> [6]: Surrogate Gap Minimization Improves Sharpness-Aware Training, ICLR 2022.
>
> ---

---

### Official Review · Reviewer_WDfQ · 2025-03-11

**Overall Recommendation:** 5

**Summary:**

In the paper, authors propose a novel mixed-precision quantization method (ASGA) via learning the sharpness of  loss landscapes, which improves the quantization generalization across datasets, thereby reducing the search cost. Particularly, the idea of introducing sharpness measure into quantization is interesting, and shows an obvious merit: the quantization policy searched from the small dataset can be directly implemented in the large one. To realize this, authors propose an enhanced sharpness measure, which aligns gradient directions, and dynamically adjusts the perturbation radius during training. This way can well deal with conflicts between different optimization objectives, and accelerate the optimization convergence. Also, authors provide theoretical analysis demonstrating the effectiveness of ASGA in reducing the upper bound of quantization generalization error and ensuring convergence. Experimental results show that ASGA achieves significant speedup in policy search compared to state-of-the-art methods while keeping comparable accuracy on large datasets.

**Claims And Evidence:**

The paper conducts both theoretical analysis and extensive experiments on a variety of datasets and models. The results demonstrate that the proposed method can achieve high accuracy with less search cost, validating the claims.

**Essential References Not Discussed:**

The paper comprehensively covers relevant MPQ and sharpness-related literature.

**Experimental Designs Or Analyses:**

The use of multiple proxy and target datasets in the experiments is valid as it helps assess generalization.

**Methods And Evaluation Criteria:**

The Adaptive Sharpness-Aware Gradient Aligning (ASGA) method is designed to tackle the challenge of heavy search overhead in Mixed-Precision Quantization (MPQ). It is reasonable to utilize proxy datasets such as CIFAR10, along with benchmark datasets like ImageNet and VOC, to assess the generalization capabilities and performance of quantization strategies across different tasks.

**Other Comments Or Suggestions:**

Please check Other Strengths And Weaknesses.

**Other Strengths And Weaknesses:**

Strengths
1. The idea of the work is interesting. The use of loss-sharpness information in transferable quantization search (which can be trained with small proxy datasets) is well-motivated, showing great potential to deal with the efficiency issue suffered by current MPQ methods.
2. The paper is well-organized and the presentation is generally clear.
3. The thermotical analysis from Lemma 1 is important, showing the  impact of loss sharpness on the upper bound of quantization generalization error.
4. The observation about the relationship between sharpness and MPQ transferability is important for understanding the motivation of the work.
5. The strategies tailored for gradient direction alignment is clear and also effective.
6. The experiments are convincing, which demonstrates that a small sharpness measure learned from proxy dataset helps seek a transferable MPQ policy for quantizing the model trained on large datasets. The experiment results are good both on the accuracy and efficiency.
7. The experimental results for ResNet18 and ResNet50 on CIFAR10 are promising. The proposed method achieves considerable speedup compared  to SOTA methods.
8. Also, the method has great potential for practical applications of neural network quantization.


Weaknesses
1. The paper employs certain assumptions in its theoretical analysis, such as those related to PAC-Bayesian theory and assumptions in the convergence analysis of stochastic optimization. These assumptions may not always hold true in practical application scenarios. Authors should explain how the theoretical conclusions drawn from these assumptions effectively guide the actual mixed-precision quantization (MPQ) tasks.
2. In the experimental section, you selected CIFAR10, Flowers, and Food as proxy datasets, and used ImageNet and VOC as target datasets. Authors should ensure that this choice of datasets adequately represents a wide range of scenarios and confirm that the performance of the ASGA method still holds for other types of datasets. Additionally, authors should have plans to conduct experiments on a broader range of datasets for validation.
3. The paper mentions that the ASGA method has been tested on different models, such as ResNet-18, ResNet-50, and MobileNet-V2. However, these models have significantly different structural characteristics. The authors should clarify the specific performance of the ASGA method in terms of its adaptability to model structures. Additionally, the authors should also consider whether the method needs to be adjusted accordingly for more complex or specialized model structures.
4. This paper mainly focuses on image classification and object detection tasks in the theoretical analysis and experimental verification. It is recommended that the authors clarify the applicability of the ASGA method in other deep learning fields, such as natural language processing and speech recognition. The authors should also determine whether this method requires significant modification to be extended to these fields.

**Questions For Authors:**

Please check Other Strengths And Weaknesses.

**Relation To Broader Scientific Literature:**

The paper's key contribution of using loss landscape sharpness for MPQ policy search builds on prior work on sharpness and generalization in neural networks. It differentiates from existing MPQ methods which rely on complex feature map calculations. By leveraging simple sharpness measures, it offers a more efficient approach, filling a gap in the literature where no prior work applied sharpness for MPQ generalization.

**Theoretical Claims:**

The proof of generalization performance in Lemma 1 is clear, which is based on PAC–Bayesian theory. The convergence analysis in Lemma 2 is correct, with clear steps.

---

> ### Author Rebuttal · Authors · 2025-03-31
>
> ## Q1 The rationality of the theoretical assumptions in the article.
>
> A1: Many thanks for your insightful comments! As you mentioned, our theoretical analysis employs certain assumptions, including the PAC-Bayesian framework and convergence analysis under stochastic optimization. We adopt PAC-Bayesian theory primarily because it provides a rigorous theoretical foundation for model generalization capability. Building upon this framework, we theoretically demonstrate that reduced loss sharpness in quantized models leads to tighter upper bounds on generalization error, thereby enhancing the generalization of MPQ policies. Our experimental results validate this theoretical insight: when applying ASGA for MPQ search, we observe a **1.5% improvement in Top-1 accuracy on ImageNet compared to conventional methods**. This empirically confirms that PAC-Bayesian error bounds effectively measure MPQ generalization capability in practice. Furthermore, **Figure 6 in our paper** illustrates the sharpness reduction during MPQ search, demonstrating ASGA's effectiveness in minimizing $\sigma_{max}$. These experimental findings align precisely with our theoretical predictions, reinforcing confidence in the validity of our analysis. Additionally, we analyze ASGA's convergence properties and derive its convergence upper bound under non-convex optimization conditions. As evidenced in **Figure 4**, our method demonstrates **stable convergence**, where the adaptive $\rho$ parameter **enhances training stability** for MPQ.
>
> ## Q2 Is the dataset used representative?
>
> A2: Many thanks. We selected multiple proxy and target datasets based on their widespread use and representativeness in image classification tasks. These datasets cover diverse image categories, styles, and difficulty levels, providing preliminary validation of ASGA's effectiveness and generalization capability. Experimental results demonstrate stable improvements in MPQ transfer performance across all scenarios. Although these rigorously selected datasets can, to a certain extent, already reflect the performance of ASGA quite well, it is undeniable that they still have certain limitations. For example, the data coverage in some specific fields may not be comprehensive enough, and so on, we believe ASGA's theoretical foundation and design approach possess universal applicability. To further evaluate ASGA's broad applicability, we plan to test it on larger-scale datasets, non-vision tasks (speech classification, autonomous driving detection), and extremely low-bit quantization tasks to ensure its effectiveness in more complex quantization environments.
>
> ## Q3 The adaptability and scalability of the proposed method to models?
>
> A3: Thanks for your comment! ASGA **requires no substantial modifications when applied to different models**. The core idea of the ASGA method lies in smoothing and optimizing the loss landscape. Through this approach, it effectively reduces the undesirable characteristics of the loss landscape, such as sharp local minima, and thereby significantly enhances the model's generalization ability, enabling it to better handle different input data and task scenarios. Our experimental validation in **Table 1 in Section 3 of our paper** across diverse model architectures, including ResNet18 and others, has consistently demonstrated improved performance, confirming ASGA's applicability to various structural designs. While minor parameter adjustments might be necessary when applying ASGA to more complex or specialized model architectures, such potential modifications would not compromise the method's general applicability. The architectural independence of our approach stems from its operation at the fundamental level of loss landscape optimization rather than specific network topologies.
>
> ## Q4 The scalability of the proposed method in other deep learning fields.
>
> A4: Thanks. As previously mentioned, ASGA demonstrates considerable versatility and can be applied to other deep learning domains without requiring substantial modifications. We plan to not only further investigate ASGA's performance in areas such as natural language processing, where its ability to handle sparse gradients and non-convex landscapes could offer distinct advantages, but also to deploy ASGA across diverse hardware scenarios, including both low-power devices and high-throughput systems, to explore its broader potential applications. These efforts will be complemented by rigorous benchmarking against state-of-the-art optimizers to obtain trade-offs in convergence speed, memory efficiency, and generalization.

---

> > ### Comment · Reviewer_WDfQ · 2025-04-03
> >
> > I have carefully reviewed the authors’ rebuttal, and all of my previous questions have been clearly addressed. I have updated my evaluation accordingly.

---

### Official Review · Reviewer_XyLq · 2025-03-12

**Overall Recommendation:** 4

**Summary:**

This paper proposes an Adaptive Sharpness-Aware Gradient Aligning (ASGA) method for generalizable mixed-precision quantization. ASGA aims to address the issue of excessive search burden in MPQ through quantization generalization, i.e., searching for quantization policies on small proxy datasets and then generalizing them to the large-scale dataset tasks. Specifically, authors consider a series of sharpness improvement strategies, such as SAM, gradient alignment, and adaptive perturbation radius strategy, which can balance accuracy, complexity, and generalization in the MPQ search. The paper conducts both theoretical analysis and experiments on a set of datasets, which have validated the effectiveness of the proposed method. Ablation studies confirm the effectiveness of its components. In summary, this work provides a new way to address MPQ's high search burden.

## update after rebuttal

**Claims And Evidence:**

In this work, all the statements are strongly backed by theoretical demonstrations or experimental findings, and all of them are without problems.

**Essential References Not Discussed:**

The related works have discussed the most relevant works about MPQ and Sharpness.

**Experimental Designs Or Analyses:**

The datasets and baseline methods used in the article are representative and can be used to evaluate the effectiveness of the method.

**Methods And Evaluation Criteria:**

ASGA aims to solve the high search cost in MPQ. Using proxy datasets like CIFAR10 and benchmark datasets such as ImageNet and VOC helps evaluate the generalization and performance of quantization strategies for various tasks.

**Other Comments Or Suggestions:**

Please refer to the weaknesses presented in Section “Other Strengths and Weaknesses”.

**Other Strengths And Weaknesses:**

Strengths

1. The paper has a clear structure and its idea is well illustrated, which is very easy to understand.

2. The utilization of the sharpness of the loss landscape for MPQ policy search is novel.

3. This design has sufficient theoretical proofs as support.

4. The experimental design of the work is reasonable and highly persuasive, which can verify the effectiveness of the proposed method.

5. The proposed method is computationally efficient since it does not introduce additional computational resource overhead.

6. The design about how to exploit the loss-sharpness information is clear.

7. The experimental results show the effectiveness of the proposed method in tackling various MPQ tasks.

Weaknesses

1. The authors should briefly explain why σmax can measure the sharpness of the loss landscape.

2. Authors enhance model generalization by minimizing the loss sharpness. However, during actual training, the model may get stuck in locally flat regions, leading to convergence to suboptimal solutions. How can we ensure that the ASGA method, while pursuing a flat loss landscape, does not excessively get trapped in locally flat regions and miss the global optimum solution?

3. Has the author considered extending the ASGA method to other fields? For example, applying the ASGA method to other types of quantization methods, such as binary quantization?

4. In the paper, authors  claim that the ASGA method  can be applied to edge intelligence applications. A natural question is that whether  the ASGA can be applied to other kinds of hardware platforms. For example, deploying this method on a Raspberry Pi?

**Questions For Authors:**

Please refer to the weaknesses presented in Section “Other Strengths and Weaknesses”.

**Relation To Broader Scientific Literature:**

The paper innovatively uses loss landscape sharpness for enhancing MPQ policy search, which is built on neural network sharpness-generalization research. Unlike existing MPQ methods with complex feature map calculations, it uses simple sharpness measures for a more efficient approach. This work is the first attempt to apply the sharpness of the loss landscape to the MPQ generalization.

**Theoretical Claims:**

I have carefully reviewed the theoretical proof sections in the article and believe they are correct.

---

> ### Author Rebuttal · Authors · 2025-03-31
>
> ## Q1 The authors should briefly explain why σmax can measure the sharpness of the loss landscape.
>
> A1: Many thanks for your comments! First, reference [1] has proven that $\sigma_{max}$ is positively correlated with the sharpness of the loss landscape. Moreover, $\sigma_{max}$ is the eigenvalue with the largest absolute value of Hessian, which reflects the curvature information of the loss landscape near a local minimum. In other words, a larger curvature indicates a sharper loss surface, leading to poorer model generalization. Conversely, smaller $\sigma_{max}$ corresponds to flatter regions where the loss changes gradually, leading to better generalization. Therefore, using $\sigma_{max}$ as a measure of the sharpness of the loss surface is appropriate.
>
> ---
>
> [1]: Surrogate Gap Minimization Improves Sharpness-Aware Training, ICLR 2022.
>
> ---
>
> ## Q2 Authors enhance model generalization by minimizing the loss sharpness. However, during actual training, the model may get stuck in locally flat regions, leading to convergence to suboptimal solutions. How can we ensure that the ASGA method, while pursuing a flat loss landscape, does not excessively get trapped in locally flat regions and miss the global optimum solution?
>
> A2: Thanks for the insightful comment! Indeed, avoiding convergence to flat regions while pursuing a flat loss landscape is a critical challenge. We recognize this issue, and thus incorporates both empirical loss and perturbed loss into the design of the ASGA loss function. This aims to maintain low loss value while seeking a flat landscape. In addition, the implicit gradient alignment mechanism helps minimize sharpness while preventing convergence stagnation caused by gradient conflict. This ensures that the model converges along a flat yet globally optimal direction guided by the loss landscape. From an experimental perspective, we validate our strategy across multiple datasets, demonstrating that our method can **obtain superior quantization policies to direct optimization on the target dataset while simultaneously improving generalization performance**.
>
> ## Q3 Has the author considered extending the ASGA method to other fields? For example, applying the ASGA method to other types of quantization methods, such as binary quantization?
>
> A3: Many thanks. We are indeed considering applying ASGA to other quantization methods. The core idea of ASGA - improving model generalization by optimizing the sharpness of the loss landscape - **is fundamentally generalizable**.  For instance, the binary quantization you mentioned represents an extreme form of low-bit quantization. However, this method has very strong constraints, which may lead to a drastic change in the loss landscape. As a result, the model optimization process becomes extremely difficult, and it is challenging to find the optimal solution. **ASGA's sharpness optimization and adaptive gradient alignment mechanism can effectively mitigate this issue**. Therefore, we plan to further investigate and validate the effectiveness of ASGA in other quantization approaches in our future research. We are confident that ASGA can also demonstrate excellent performance in other quantization methods.
>
> ## Q4 In the paper, authors claim that the ASGA method can be applied to edge intelligence applications. A natural question is that whether the ASGA can be applied to other kinds of hardware platforms. For example, deploying this method on a Raspberry Pi?
>
> A4: Thanks for the comment! In edge intelligence applications, the primary challenges are to reduce computational overhead and enhance model generalization—objectives that align closely with ASGA’s design goals. While this paper focuses on ASGA’s application to conventional edge devices, we believe our method can be applicable to resource-constrained platforms such as Raspberry Pi. As pointed out in Section 3 of our paper, unlike the traditional model training process, ASGA only needs to conduct policy search on a small-scale proxy dataset and then finetune it on the large-scale target dataset. This approach can **significantly reduce a large amount of computational overhead for resource-constrained devices**. Thus, even for low-power devices like the Raspberry Pi, ASGA can be practically deployed by first performing offline quantization policy searches before on-device deployment, making it feasible for energy-efficient scenarios.

---

### Decision · Program_Chairs · 2025-05-01

**Decision:**

Accept (poster)

**Comment:**

This paper tackles the high computational cost of Mixed-Precision Quantization (MPQ) policy search for compressing DNNs, which traditionally requires searching on the large target dataset. The authors propose searching on a small proxy dataset instead, but this usually suffers from poor generalization due to data distribution shifts. To fix that, they leverage the link between flatter loss landscapes and better generalization. Specifically, they introduce a method that optimizes the MPQ policy search on the proxy dataset to favor solutions leading to flatter loss minima for the quantized model. This enhances policy transferability without needing complex feature analysis like prior methods. Experiments confirm finding effective and transferable MPQ policies using very small proxy datasets (e.g., 0.5% of target size), significantly accelerating the search process (113-150% speedups over SOTA) while maintaining good performance on the final model.

The reviews favor acceptance due to a successful author rebuttal addressing initial concerns. Reviewers generally recognized the value and novelty of the proposed method to drastically reduce MPQ policy search costs. Highlighted strengths include the intriguing use of loss sharpness ideas for MPQ transferability, the method's significant potential for improving search efficiency, its theoretical underpinnings, and strong experimental results demonstrating speedups and competitive accuracy. However, initial reviews also raised several important questions regarding the distinctiveness from prior sharpness-aware work (like SAQ), the completeness and fairness of experimental comparisons (especially against baselines like HAWQ and the use of convergence epochs versus wall-clock time), the practical scope across different datasets and models, and clarity on some technical aspects.

The authors provided a detailed rebuttal that effectively tackled these critical points. They successfully differentiated ASGA from related work by clarifying its specific goals within the cross-dataset MPQ context. The authors also provided additional experimental evidence, including comparisons using GPU hours which validated the claimed efficiency gains, and demonstrated how ASGA could complement strong baselines like HAWQ, particularly when dealing with dataset distribution shifts. They also justified their evaluation metrics within their proposed two-stage workflow and provided requested technical clarifications regarding the sharpness measure and optimization details. This comprehensive response alleviated the main reservations by the reviewers, leading to positive final recommendations, with one reviewer explicitly increasing their score citing the clarified benefits of ASGA for model calibration across datasets.

In agreement with the reviewers, I find that this submission successfully demonstrates the benefits of leveraging loss sharpness in the novel application domain of Mixed-Precision Quantization (MPQ) to propose a new algorithm, and supports that by strong and promising empirical results. I recommend acceptance.